Differentially expressed genes and key molecules of BRCA1/2-mutant breast cancer: evidence from bioinformatics analyses

Li Yue 1
Zhou Xiaoyan 1
Liu Jiali 1
Yin Yang 1 2
Yuan Xiaohong 1
Yang Ruihua 1
Wang Qi 1
Ji Jing 1
He Qian qianh0511@126.com 1
1 Department of Clinical Laboratories, Second Affiliated Hospital of Xi’an Jiaotong University , Xi’an , China
2 Department of Clinical Laboratories, XIAN XD Group Hospital , Xi’an , China
Uversky Vladimir
Electronic publication date: 2020 Jan 21
Publication date: 2020
Volume: 8
Electronic Location ID: e8403
Received 2019 Oct 1; Accepted 2019 Dec 16
Copyright: ©2020 Li et al.
Copyright year: 2020
Copyright holder: Li et al.
License: This is an open access article distributed under the terms of the Creative Commons Attribution License, which permits unrestricted use, distribution, reproduction and adaptation in any medium and for any purpose provided that it is properly attributed. For attribution, the original author(s), title, publication source (PeerJ) and either DOI or URL of the article must be cited.
License URL: https://creativecommons.org/licenses/by/4.0/

Keywords: Breast cancer, BRCA1/2 mutations, Differentially expressed genes, Survival analysis, diagnostic value

Funding: Natural Science Basic Research Program of Shaanxi province and National Natural Science Foundation of China 8160031466 The study is supported by Natural Science Basic Research Program of Shaanxi province and the National Natural Science Foundation of China (grant no. 8160031466). The funders had no role in study design, data collection and analysis, decision to publish, or preparation of the manuscript.

==============================
Background

BRCA1 and BRCA2 genes are currently proven to be closely related to high lifetime risks of breast cancer. To date, the closely related genes to BRCA1/2 mutations in breast cancer remains to be fully elucidated. This study aims to identify the gene expression profiles and interaction networks influenced by BRCA1/2 mutations, so as to reflect underlying disease mechanisms and provide new biomarkers for breast cancer diagnosis or prognosis.

Methods

Gene expression profiles from The Cancer Genome Atlas (TCGA) database were downloaded and combined with cBioPortal website to identify exact breast cancer patients with BRCA1/2 mutations. Gene set enrichment analysis (GSEA) was used to analyze some enriched pathways and biological processes associated BRCA mutations. For BRCA1/2-mutant breast cancer, wild-type breast cancer and corresponding normal tissues, three independent differentially expressed genes (DEGs) analysis were performed to validate potential hub genes with each other. Protein–protein interaction (PPI) networks, survival analysis and diagnostic value assessment helped identify key genes associated with BRCA1/2 mutations.

Results

The regulation process of cell cycle was significantly enriched in mutant group compared with wild-type group. A total of 294 genes were identified after analysis of DEGs between mutant patients and wild-type patients. Interestingly, by the other two comparisons, we identified 43 overlapping genes that not only significantly expressed in wild-type breast cancer patients relative to normal tissues, but more significantly expressed in BRCA1/2-mutant breast patients. Based on the STRING database and cytoscape software, we constructed a PPI network using 294 DEGs. Through topological analysis scores of the PPI network and 43 overlapping genes, we sought to select some genes, thereby using survival analysis and diagnostic value assessment to identify key genes pertaining to BRCA1/2-mutant breast cancer. CCNE1, NPBWR1, A2ML1, EXO1 and TTK displayed good prognostic/diagnostic value for breast cancer and BRCA1/2-mutant breast cancer.

Conclusion

Our research provides comprehensive and new insights for the identification of biomarkers connected with BRCA mutations, availing diagnosis and treatment of breast cancer and BRCA1/2-mutant breast cancer patients.

Introduction

Breast cancer susceptibility gene (BRCA1 and BRCA2) mutations, which confer substantial lifetime risks of breast and ovarian cancers (Atchley et al., 2008), influence oncogenesis and metastasis of breast cancer (Kuchenbaecker et al., 2017). BRCA1/2 are currently proven to be closely related to hereditary breast cancer and some sporadic breast cancer. But there is a paucity of data pertaining to ethnical high-risk cases with BRCA1/2 mutations and further large BRCA mutation prevalence studies (Bernstein-Molho et al., 2019; Armstrong et al., 2019). Although some genes have been identified and the pathogenic mechanism of BRCA1/2 genes for breast cancer has partly explained, the closely related genes to BRCA1/2 in breast cancer (BC) remain to be fully elucidated.

The identification of BRCA1/2 mutation carriers only relies on, genetic testing for high-risk patients judged by their information that have family history or initial clinical symptoms (Foulkes, 2013; Shimada et al., 2019). In fact, this also limits the opportunity of prevention for BRCA1/2-mutant breast cancer and other tumors such as ovarian cancer, due to the cost effectiveness for extending to population-based sequencing (sequencing costs not offset by healthcare benefits of preventing future malignancies) (Gourley, 2019) and limitations of BRCA gene mutation detection. Loss of one copy of functional BRCA1/2 is not clinically apparent, and somatic mutations detection of BRCA genes is affected by cancer cell content and mutation ratio, lacking the accuracy and inherent simplicity, and the accuracy of detection. Although germline and somatic variants of BRCA1/2 have been described, variants in their genetic regions only account for a small proportion of cancer risk, and the majority is currently unknown, which remains a difficulty for genetic testing (Santana Dos Santos et al., 2018). Moreover, BRCA1/2 mutations render tumors more sensitive to drugs that cause DNA cross-linking, such as cisplatin, carboplatin, and mitomycin. In clinical practices, PARP1 inhibitors, represented by olaparib, have become monotherapies for patients with BRCA-mutant cancer (Tutuncuoglu & Krogan, 2019), but perhaps inevitably, long-term effectiveness of which is hindered by their progressive resistance (Barber et al., 2013; Macedo & Ashton-Prolla, 2019). Due to the difficulty in identifying and treating BRCA1/2-mutant BC, it is of great importance to find more key candidate genes for the diagnosis and treatment of BC, especially for some hereditary and sporadic BC, and understand underlying pathogenesis mechanisms of BRCA mutations.

In recent years, large-scale genome sequencing, such as high-throughput data including The Cancer Genome Atlas (TCGA) database, provides a new method to help researchers explore the complex relationship between genetic molecules and disease (Huang & Li, 2017; Zhai et al., 2019). So, in this study, we screened the transcriptome sequencing dataset of appropriate BRCA mutant and wild-type BC patients from the TCGA database, and thereby identified differentially expressed genes (DEGs) through analysis of these two sets of data to reflect gene expression profiles influenced by BRCA1/2 mutations, combined with Gene set enrichment analysis (GSEA), survival analysis and diagnostic value assessment. Protein–protein interaction (PPI), survival analysis and diagnostic value assessment help us identify key genes associated with BRCA1/2 mutations and provide new insights for the specific mechanisms and treatment targets research of BRCA-mutant breast cancer different from other breast cancers.

Material and Methods

RNA-seq data

An RNA-Seq dataset of breast cancer, which included the whole transcriptome sequencing dataset and corresponding clinical profiles of over 1000 human BC patients, was download from TCGA database (https://portal.gdc.cancer.gov/). The format of mRNA-seq data is HTseq-Counts which can be analyzed for differential gene expression using the edgeR package, and HTseq-FPKM for functional annotation, pathway enrichment and diagnostic value. The corresponding information related to patients with BRCA1/2 mutations (MUT) was obtained from the cBioPortal website (http://www.cbioportal.org/index.do) (Gao et al., 2013), including mutation and copy number variation (CNV), in order to create MUT group satisfying BRCA1 or BRCA2 mutation with complete RNA-seq data and clinical data. The BRCA1/2 wild-type (WT) group was randomly selected without mutation from all breast cancer RNA-seq data, and had complete RNA-seq data and clinical data. Moreover, we chose all correspondent para-carcinoma tissue samples from BC RNA-seq data as control group, and the total number is 112. The overall schematic of methods used in this study is shown in Fig. 1.

Figure 1 Flow chart of methodologies used in this study.

Note: The BRCA1/2-mutant (MUT) group was set following the inclusion and exclusion criteria: (1) Data were included when (i) mutation or CNV was shown in both data sets (TCGA PanCancer Atlas and TCGA Provisional) searching by cBioPortal website, (ii) data had complete RNA-seq data and clinical data; (2) Data were directly excluded when amplification was detected in any data set for corresponding samples. The BRCA1/2 wild-type (WT) group was randomly selected without mutation from all breast cancer RNA-seq data, and had complete RNA-seq data and clinical data. We also classified para-carcinoma samples as control group. And then three differentially expressed genes (DEGs) analysis were performed on three groups in pairs, namely MUT versus (vs.) WT, MUT vs. Control and WT vs. Control. This was followed by applying a PPI integration using the differentially expressed genes in MUT vs. WT as input. The DEGs results of three comparison helped us further screen and identify key candidate genes, and conduct survival analysis and evaluate the diagnostic efficacy for genes closely associated with BRCA1/2 mutations.

Gene set enrichment analysis (GSEA)

To study the effect of BRCA1/2 mutations on various biological function gene sets in breast cancer patients, GSEA was adopted to analyze the differences of two groups (with or without BRCA mutations) in gene mRNA expression levels of biological functional annotation and pathways. Hereby, the number of permutations was set at 1,000, and the remaining were default parameters. Reference gene sets database from Molecular Signatures Database (MSigDB) of h(h.all.v6.2.symbols.gmt), c2 (c2.cp.kegg.v6.2.symbols.gmt) and c5 (c5.bp.v6.2.symbols.gmt; c5.mf.v6.2.symbols.gmt; c5.cc.v6.2.symbols.gmt; consist of genes annotated by the same GO terms), respectively (Liberzon et al., 2015). The MSigDB of h was a hallmark gene sets, constructed on marker genes associated with various cellular biological processes including cell apoptotic and division; c2 was a pathway gene set, which was curated from publications and extracted from canonical pathways and experimental signatures; c5 was Gene Ontology(GO) gene sets, consisted of biological process(BP), cellular component and molecular function. Enrichment analysis was considered statistically significant when meeting the following criterion: nominal P-value cutoff (NOM p-value) <0.05 and false discovery rate (FDR) <0.25.

Identification of differential gene expression (DEGs)

Expression profile data of the analyzed groups in the study (MUT vs WT; MUT vs control; WT vs control), were managed by gene ID conversion and default value removal using R package. A total of 19611 genes per sample were available for analysis in the matrix file. EdgeR, an R package for examining DEGs of RNA-Seq count data, was used three times alone without interference, to identify differentially expressed genes between BRCA1/2-mutant (MUT) BC patients and wild-type (WT) BC patients, between MUT and control samples, between WT BC and control samples, respectively. Differentially expressed genes were corrected by FDR adjustment and considered significant following the criterion: —log2 fold change (FC)— ≥1; both the P-value and FDR<0.05. We mainly used differentially expressed genes between BRCA1/2- mutant BC patients and wild-type BC patients to conduct further analysis, including functional annotation, pathway enrichment and PPI analysis. Moreover, survival analysis and diagnostic efficacy were performed, based on the identification of more meaning genes which were considered differentially expressed in all three comparisons.

Protein–protein interaction (PPI) network construction

For the DEGs between BRCA1/2-mutant BC patients and wild-type BC patients, PPI construction helped our understand relationships of these genetic expression changes, closely related to BRCA1/2 mutations. We performed PPI network by STRING database (https://string-db.org/), a common online approach known to predict protein-protein interactions, followed by PPI network visualization in Cytoscape (V.3.7.0). Based on the results from the STRING database and analysis from Cytoscape and its plug-in cytohubba, we synthetically evaluated all the genes by 12 topological analysis methods including Degree, Clustering Coefficient and so on, provided by cytohubba (Chin et al., 2014), to identify some specific hub genes closely related to BRCA mutations.

Further analysis

Key candidate genes which were all considered differentially expressed in all three comparisons, were screened to evaluate their prognosis and diagnosis information for breast cancer and BRCA-mutant breast cancer. For this purpose, we used Kaplan–Meier plotter (http://kmplot.com/analysis/), a software available online that specializes for survival analysis (Gyorffy et al., 2010). Herein, the overall survivals (OS) of BC patients were analyzed using the Kaplan–Meier method, based on the classification where patients were divided into a high and low genetic expression group according to the expression level of genes. Survival analysis was considered statistically significant while P < 0.05.

The receiver operating characteristic curve (ROC) was used to evaluate the diagnostic efficacy of the indicators and to calculate the area under the curve (AUC) by SPSS 18.0. Next, we analyzed their mRNA expression levels using the GraphPad Prism 7 software combined the corresponding mRNA-seq data of three groups in this case, and also examined the expression of candidate genes in ethnic sub-division of three groups to reflect the potential effects of race on the final results.

Results

Data source

Through TCGA database, we obtained complete BC clinical profiles and corresponding RNA-Seq dataset. There were about 7–10% BC patients with BRCA1/2 mutations and the rest were BRCA1/2 wild type, obtained from the cBioPortal website; among them, the proportion of BRCA1 &BRCA2 mutations was 3% (38/1094, TCGA: Provisional) & 4% (48/1094, TCGA Provisional) (or, 4%,45/1084 & 5%,54/1084, in TCGA: PanCancer Atlas, shown in Fig. S1A). The main mutation type was missense mutation and truncating mutation in BRCA1 and BRCA2 genes (Fig. S1B). If only choosing cases with demonstrated mutation in both datasets (TCGA Provisional & TCGA: PanCancer Atlas) and satisfying group criteria, we finally confirmed 42 mutant cases. Demographic data between MUT and WT groups are presented in Table 1.

Table 1 Clinical characteristics data for main subjects used in this study.

	MUT group (BRCA1/2-mutant BC tissue)	WT group (randomly selected BC tissue without BRCA1/2 mutations)	
N	42	117	
Age (initial diagnosis)	57.2 ±13.2	58.3 ±13.2	
Race			
White	30/36 (83.3%)	83/109 (76.5%)	
African	4/36 (11.1%)	19/109 (17.4%)	
Asian	2/36 (5.6%)	7/109 (6.4%)	
Stage			
Stage I	4/42 (9.5%)	21/117 (17.9%)	
Stage II	32/42 (76.2%)	65/117 (55.6%)	
Stage III	6/42 (14.3%)	28/117 (23.9%)	
Stage IV	0	3/117 (2.6%)	
Immune phenotype			
ER −	18/42 (42.9%)	26/110 (23.6%)	
PR −	23/42 (54.8%)	35/109 (32.1%)	
HER2			
amplifications	11/39 (28.2%)	15/97 (15.5%)	
HER2 −	28/39 (71.8%)	82/97 (84.5%)	

GSEA Enrichment analysis

To investigate the effect of BRCA1/2 mutations on progression and prognosis of breast cancer, the influences of biological functional annotation sets were analyzed by GSEA method (shown in Fig. 2). The seven, eighty-three or one consensus gene sets from Hallmark collection, c2 KEGG-sub collection or c5 collection, respectively, were significantly enriched in MUT group compared with WT group. Among these enrichment items, gene sets associated with mitotic spindle (e.g., Fig. 2B, Fig. 2E, Fig. 2H), cell cycle (Fig. 2F, Fig. 2K), G2M checkpoint (Fig. 2A) and so on were obviously enriched. Hallmark gene and biological processes pertaining to regulation of transcription involved in G1-S transition, mitotic spindle organization, cell cycle phase transition, ATP dependent chromatin remodeling, cell cycle G1/S transition, negative regulation of cell division, cytoskeleton dependent cytokinesis, cell cycle checkpoint, E2F and MTORC1 signaling, etc., were significantly enriched, suggesting that BRCA1/2 mutations may contribute to disease progression and affect prognosis mainly by influencing cell proliferation via regulation of cell cycle, cell division and gene replication in breast cancer patients. The GO enrichment analysis of molecular function was significantly enriched in structural constituent of cytoskeleton. Furthermore, the cellular component was enriched for kinetochore, spindle midzone and so on. In the GSEA analysis of KEGG pathways, the BRCA1/2 mutation group was associated with cell cycle (Fig. 2K).

Figure 2 GSEA Enrichment analysis results of BRCA1/2 mutations in breast cancer patients.

Note: GSEA Enrichment analysis including H (figure 3A–D), c2 (K) and c5 consisted of biological processes (bp, E–J), cellular component (cc, H–I) and molecular function (mf, J). GSEA, Gene set enrichment analysis.

Identification of DEGs

Overall, RNA-Seq datasets from 42 BRCA1/2 mutation-bearing patients and randomly selected 117 wild-type BC patients were used for DEG screening. A total of 294 DEGs were identified between BRCA-mutant and wild-type BC, of which, 199 were upregulated and 95 were downregulated. Furthermore, we performed differentially expressed genes analysis between MUT and control group and identified 4851 differentially expressed genes in MUT group. In addition, comparison of WT (breast cancer) and control group (para-carcinoma tissue) identified 4990 differentially expressed genes in BC patients. The volcano plots of the DEGs were shown in Figs. 3A–3C. More importantly, Venn analysis for three comparisons emphasized a combination of 43 overlapping DEGs (Fig. 3D and Table S1), suggesting the expression of these genes not only significantly changed in breast cancer patients but more obvious significantly changed in BRCA 1/2-mutant breast patients. These genes might participate in the specific molecular mechanisms of the carcinogenesis of BRCA mutations. The top 10 upregulated and downregulated overlapping DEGs based on fold changes were listed in Table 2.

Figure 3 Volcano plot for DEGs and Venn plot of three independent DEGs identifications.

Note: As stated earlier, we performed identification of DEGs three times, between BRCA1/2-mutant (MUT) group and wild-type (WT) group, MUT and para-carcinoma (Control) group, WT and Control group, respectively. Their volcano plots were shown in figure 3A–C respectively. (D), Venn analysis of above three independent DEGs; (E), the number of differentially expressed genes in three comparisons. Blue: high expression; Yellow: low expression; Black dots: the genes with expression of —log2FC— < 1 or FDR > 0.05. LogFC, log2 fold change; DEGs, differentially expressed genes.

Table 2 The top ten genes with the most obvious expression changes, screened by the identification of DEGs in three comparisons.

Category	Gene symbol	Log FC	
		MUT vs WT	MUT vs control	WT vs control	
Top ten upregulated genes in BRCA1/2-mutant breast cancer	CT45A10	4.46	7.66	3.22	
TBX10	3.94	6.63	2.61	
NLRP7	3.26	4.97	1.71	
BARHL2	3.24	5.40	2.15	
C4orf51	2.79	4.63	1.82	
CLLU1OS	2.68	4.15	1.47	
TUBB4A	2.67	4.14	1.43	
NPBWR1	2.32	4.16	1.83	
A2ML1	2.02	4.30	2.27	
TTK	1.00	3.68	2.68	
Top ten downregulated genes in BRCA1/2-mutant breast cancer	MYOM2	−3.24	−5.64	−2.42	
CA4	−3.11	−8.48	−5.35	
LGALS12	−2.41	−6.47	−4.04	
SLC4A4	−2.07	−3.48	−1.37	
CAPN11	−1.91	−3.23	−1.36	
HPSE2	−1.90	−5.12	−3.23	
MAOA	−1.89	−4.55	−2.65	
DNAH9	−1.76	−3.21	−1.43	
RELN	−1.75	−4.40	−2.66	
NNAT	−1.55	−4.73	−3.16	

PPI network of DEGs

Altogether, 95 downregulated and 199 upregulated DEGs in MUT vs WT were submitted for further PPI network construction with STRING database and cytoscape software (version 3.7.0), to reflect the specific genetic interaction networks associated with BRCA1/2 mutations compared with WT group. A total of 209 nodes and 498 edges were mapped in the PPI network (shown in Fig. 4), with an average node degree of 3.34, average local clustering coefficient of 0.372 and a PPI enrichment P value <1.0e −16. If all the genes were synthetically evaluated by 12 topological analysis methods from plug-in cytohubba (Chin et al., 2014), we chose top five genes as hub genes for mutant group. These hub genes were serum albumin (ALB), CDKN1A (cyclin-dependent kinase inhibitor 1), CCNE1 (G1/S-specific cyclin E1), MYOM2 (myomesin-2), KRT20 (keratin 20). Among them, CCNE1 and KRT20 were of high expression in BRCA1/2-mutant breast cancer, and ALB and MYOM2 were of low expression. Their expression level changes were shown in Table 3.

Figure 4 Visualization of the protein–protein interaction (PPI) network of DEGs in MUT vs WT group, by the means of STRING and Cytoscape tool.

A total of 294 differentially expressed genes in the comparison of MUT vs. WT were submitted for PPI network construction. A total of 209 nodes and 498 edges were mapped in the PPI network. Yellow, molecules with the node degree > 11.

Survival analysis and diagnostic efficacy of hub genes

In this study, we observed CCNE1, NPBWR1 (Neuropeptides B/W receptor 1), SLC4A4 (Solute Carrier Family 4 member 4), MAOA (Monoamine oxidase A), A2ML1 (Alpha-2-macroglobulin like 1) and TTK (dual specificity protein kinase) not only significantly expressed in BRCA1/2-mutant breast cancer compared with wild-type BC and normal tissue, but also showed significant prognostic value for breast cancer (shown in Figs. 5A–5F).

Moreover, among upregulated hub genes, we found many genes displayed good diagnostic efficacy for BRCA1/2-mutant breast cancer compared to wild-type breast cancer, including CCNE1, NPBWR1, A2ML1, TTK, C4orf51 (Chromosome 4 open reading frame 51) and EXO1 (Exonuclease 1), with AUC value >0.630 and P-value <0.05. Their ROC curves were illustrated in Figs. 5G–5L.

As shown in Fig. S2, although the NPBWR1, A2ML1 and C4orf51 displayed differential expression, their expression level in breast cancer tissues is still not high (Figs. S2D–S2F), which may be due to their own expression abundance or sensitivity of the detection probe. In view of the fact that the influence of race on genes (BRCA1/2 and other all genes) is possible but still unknown, we analyzed the relative expression level of hub genes in ethnic subgroups. As shown in Figs. S2A–S2C, we thought that ethnic differences associated with it could be acceptable in general, because the changed expression level of hub genes in the ethnic sub-division of three groups is almost still significant. The lack of data from Asian patients makes it difficult for statistical analysis of Asian ethnic subgroups. Therefore, we believed CCNE1, TTK and EXO1 were remarkably overexpressed in BC tissues compared with para-carcinoma tissue, and might be promising to screen BC and further distinguish the high risks of BRCA1/2 mutations from wild-type BC, while ignoring the potential impact of the genetic background related to the race itself to some extent.

Table 3 The expression of some genes with high topological analysis score by cytoscape.

Regulated	Gene symbol	Log FC	
		MUT vs WT	MUT vs Control	WT vs Control	
upregulated	CCNE1	1.13	3.28	2.15	
KRT20	7.41	8.03		
downregulated	MYOM2	−3.24	−5.64	−2.42	
ALB	−3.79	−4.42		

Figure 5 Prognostic and diagnostic significance of identified hub genes for breast cancer.

(A–F) illustrate the prognostic value of some hub genes for breast cancer: (A) NPBWR1, (B) CCNE1, (C) TTK, (D) A2ML1, (E) MAOA and (F) SLC4A4. Their prognostic results were obtained from Kaplan–Meier plotter, using the Kaplan–Meier method with a log-tank test, and P < 0.05 was considered statistically significant. (G–L) show their ROC curve to reflect the diagnostic values of hub genes to distinguish BRCA1/2-mutant breast cancer from wild-type breast cancer. Here, we show (G) NPBWR1, (H) CCNE1, (I) TTK, (J) A2ML1, (K) C4orf51 and (L) EXO1 that were all have good diagnostic efficacy. ROC, the receiver operating characteristic curve.

Discussion

The incidence of breast cancer among female worldwide continues to rise, despite the fact that the mortality of cancer has been decreasing due to the development of efficient screening and treatment. The relationship between gene polymorphism and susceptibility to breast cancer, and the influence of multi-gene and multi-signal pathways for the progression of breast cancer have always attracted continuous interests, which will provide ideas for early screening and individualized treatment of breast cancer. Whole genome and exome sequencing including TCGA program have provided novel insights for researchers to explore the complex relationship between genetic molecules and disease, and ultimately advance the precision medicine. From a clinical perspective, precision cancer therapeutics, aims to tailor a treatment strategy to the unique genetic background of each disease, by targeting particular mutants upon exploiting their related mechanistic characterization of the genetic interactions involved in carcinogenesis, tumor progression and metastasis (Tutuncuoglu & Krogan, 2019). In our study, from this thought, we analyzed the difference in genetic expression profiles and interaction networks between mutant and wild-type by selecting transcriptome data from breast cancer patients with BRCA1/2 mutations.

Mutations in BRCA1 or BRCA2 are closely linked to familial breast and ovarian cancers. The BRCA genes are tumor suppressor genes that play many critical roles in many tumors, the most important of which is DNA damage repair, especially double-stranded DNA (dsDNA) repair. BRCA1/2 mutations occur at scattered sites, and occur missense mutations, especially which will influence those situated in exons encoding domains that interact with BRCA1-binding proteins, such as BARD1, BRIP1 and PALB2, which (along with RAD51C, RAD51D and possibly RAP80 and FAM175A, encoding Abraxas) are also breast and/or ovarian cancer susceptibility genes (Foulkes, 2013). Zhao et al. (2017) reported that BRCA1 and its interacting proteins BARD1 functioned in DNA double-strand break repair by influencing RAD51-mediated homologous DNA pairing. In the GSEA enrichment results for BRCA1/2-mutant TCGA breast cancer patients, our study also demonstrated BRCA genes might be implicated in regulation of tumor cell cycle, through the regulation of cellular component kinetochore and mitotic spindle, and the regulation of cycle-related biological processes including G1/S transition, cell cycle checkpoint and mitotic spindle organization. Through interaction network exploration, we found that CCNE1 with higher mRNA overexpression in BRCA1/2-mutant compared with wild-type BC, might play an important role in cell cycle regulation of BRCA-mutant tumors and the tumorgenesis of BRCA mutants.

BRCA genes mutations, conferring increased risks for breast and high-grade serous cancer of the gynecological tract (fallopian tube, ovary and peritoneum), are closely associated with triple negative breast cancers (TNBC) (Foulkes, 2013; Sanford et al., 2015). Among the classifications of breast cancer, TNBC, manifested as early recurrence and poor survival, does not express estrogen receptor (ER), progesterone receptor (PR), and human epidermal growth factor receptor 2 (HER2). Among patients with TNBC, the incidence of BRCA1/2 mutation is estimated to range from 11–37% (Sanford et al., 2015; Young et al., 2009). TNBC accounts for about 70% and 16–23% of BRCA1 and BRCA2 mutation carriers, respectively (Stevens & Couch, 2013), suggesting that TNBC is also inextricably linked to germline mutation in this breast cancer susceptibility gene. In our study, we also demonstrated that BRCA1/2-mutant BC exhibited higher ER receptor and PR receptor negative rates, as well as high HER2 amplification, compared with BRCA1/2-wild type BC, by analyzing demographic data from MUT and WT group.

To date, numerous genes have been found to influence the formation and progression of breast cancer, and thus act as diagnostic and therapeutic targets with clinical potentials. Although some genes have been identified and the pathogenic mechanism of BRCA1/2 genes for breast cancer has partly explained, the closely related genes to BRCA1/2 in breast cancer remain to be fully elucidated. In our DEGs analysis, the most of the DEGs (a total of 294) obtained from the comparison of BRCA1/2-mutant and wild-type breast cancer were demonstrated to be sense; of these, 146 were also differentially expressed between mutant BC and its normal tissue, with upregulated 108 genes and downregulated 38 genes. Furthermore, some genes were further verified by the identification between wild-type BC and normal tissue, which showed that a total of 43 genes had not only significant changes of expression level in BC patients but further more obvious changes in BRCA1/2-mutant BC patients. This demonstrated specific expression alternation of genes in our study, including TTK, EXO1, TICRR (TOBPPI interacting checkpoint and replication regulation) and so on, would be closely associated with BRCA1/2 mutations, providing a clue for further understanding the certain characteristic of BRCA1/2-mutant BC (for example, increased risk of distant metastasis and more aggressive nature) (Wang et al., 2018) and clarifying its pathogenesis.

More interestingly, some candidate genes displayed potential therapeutic and diagnostic value, especially for BRCA1/2-mutant breast cancer. For example, upregulated CCNE1, TTK and EXO1 displayed good diagnostic efficacy for screening breast cancer and BRCA1/2-mutant breast cancer. We noted that the BRCA1/2 mutations rate of the White is a bit higher in WT group than that in MUT group, which might influence the reliability of our results and become an existing problem. TCGA data is mainly derived from White patients, so it could be difficult to achieve strict inter-group ethnic balance. However, in view of possible ethnical risks with BRCA1/2 (De Bruin et al., 2012), by analyzing the relative expression level of hub genes in ethnic subgroups, we thought that the potential impact of ethnic differences could be acceptable in general. Of course, it was important to note that the differences in gene expression of some genes in the Asian subgroup are sometimes insignificant, which was be due to the lack of Asian patients’ samples in TCGA data itself. Through Kaplan–Meier approach, some differentially expressed genes with good PPI network scores were evaluated their prognosis information. Herein, we only analyzed the prognostic value of these genes for the whole breast cancer due to lack of adequate BRCA1/2-mutant cases for accurate survival analysis, which was an undeniable limitation in our study. Using the survival analysis, we found that CCNE1, NPBWR1, A2ML1 and TTK might act critical functions in the oncogenesis and progression of BRCA1/2-mutant breast cancer, reflected by their diagnostic efficacy for BRCA1/2 mutations and prognostic value as well.

EXO1, a DNA mismatch repair gene, its polymorphisms have been reported to play a critical role in the development of many tumors (Shi et al., 2017; Zhang et al., 2016). Also, due to its role in DNA replication repair and homology-directed repair, the relationships of EXO1 and BRCA1/2 mutations and its underlying mechanism have become an important focus to be studied (Lemacon et al., 2017). A recent study found that the EXO1 expression level was elevated in hepatocellular carcinoma patients and its overexpression was correlated with larger tumor size, increased lymph node metastasis, and thus proving its potential therapeutic value for hepatocellular carcinoma as a promising prognostic marker (Dai et al., 2018). Our findings were validated by another study to some extent, and they found DEPDC1, EXO1, RRM2 and some proteins had enhanced expression in the ductal carcinoma in situ and invasive ductal carcinoma (Kretschmer et al., 2011). In a word, the functions of EXO1 still require further research to fully illuminate its role in the progression of BC and the carcinogenesis of BRCA1/2 mutations.

The protein encoded by CCNE1 is G1-S specific cyclin that plays an important role in regulating the transition of G1 to S cell cycle phase by binding to and activating the expression of cyclin-dependent protein kinase 2 (Cdk2) (Bendris & Blanchard, 2015). CCNE1 also has a direct role in triggering DNA replication and maintaining genomic stability. Amplification or upregulated expression of CCNE1 is associated with poor prognosis in some tumors such as breast or ovarian cancer (Karst et al., 2014; Zhao et al., 2019). In our analysis, CCNE1 was significantly upregulated in BRCA1/2-mutant BC, compared with wild-type BC, suggesting that BRCA1/2 genes could regulate cell cycle in tumors via CCNE1 reflected by the fact that cell cycle phase transition, especially cell cycle G1/S transition were significantly enriched in mutant breast cancer from the GSEA results.

NPBW1 or GPR7 (namely Neuropeptides B/W receptor 1), a protein encoded by the NPBWR1 gene, can mainly regulate physiological responses related to the nervous system, including stress response and pain response (Nagata-Kuroiwa et al., 2011). However, the specific mechanism of NPBW1 in tumorigenesis has not been studied and confirmed. Cottrell S et al. reported that methylation of GPR7 was significantly associated with prostate cancer prognosis, and would result in more accurate prediction for prostate cancer recurrence in clinical practice (Cottrell et al., 2007). A2ML1 is a broad protease inhibitor from the alpha-macroglobulin superfamily, with a unique mechanism where A2ML1 undergoes a conformational change following its cleavage by a protease and thereby traps the protease to prevent proteases from binding to their substrates (Galliano et al., 2006; Vissers et al., 2015). The clinical significance of A2ML1 has been demonstrated in paraneoplastic pemphigus (PNP), an autoimmune bullous disease accompanied by a variety of benign or malignant tumors including non-Hodgkin lymphoma (Mimouni et al., 2002). A2ML1 could serve as a useful diagnostic biomarker for PNP (Ohzono et al., 2015). Recently, a bioinformatics study pointed out the new use of A2ML1 as a diagnostic target for lung cancer (Zhang et al., 2018). In our study, we found that NPBWR1 and A2ML1 have certain prognostic or diagnostic significance for breast cancer, and we thought that the two molecules also deserve further investigated, though their expression level in breast cancer tissues is not high enough overall.

At present, many studies have found that high expression of dual specificity protein kinase (TTK), encoded by the TTK gene, is associated with the oncogenesis, progression and treatment resistance of breast cancer (especially TNBC) (Riggs et al., 2017). It was reported that TTK could regulate the growth and epithelial-to-mesenchymal transition of TNBC cells through TGF- β and KLF5 signal pathways, thereby affecting the invasion and metastasis of tumors (King et al., 2018). More future research about its mechanisms in TNBC will provide a theoretical basis for TTK inhibitor-targeted therapy in the field of breast cancer and TNBC. In the present study, the upregulated expression and corresponding diagnostic value of TTK in BRCA1/2-mutant breast cancer suggested that the role of TTK in this type of breast cancer is equally noteworthy.

Here, by many bioinformatic analyses, we identified some important molecules affected by BRCA genes mutations. Survival and diagnosis analysis, and the validation of genes in the ethnic sub-groups implied their potentials as reliable prognostic or diagnostic indicators and as possible therapeutic targets. In this article, from a new perspective, we identified some novel DEGs, including CCNE1, NPBWR1, A2ML1, EXO1 and TTK, might play critical functions in the oncogenesis and progression of BRCA1/2-mutant BC, which was not been previously interpreted from a similar idea. Of course, it is necessary to indicate that BRCA1/2-mutant patients in this study are derived from the gene mutations detection of TCGA cancer patients, who are not distinguished between somatic mutations and germline mutations (not necessarily as hereditary breast cancer patients). Moreover, in order to select more credible key candidate genes, the selection of the hub genes was validated in other comparisons of mutant and normal tissues, and of wild-type and normal tissues. In a word, our study will help explain the underlying mechanisms of BRCA in carcinogenesis, identify novel diagnostic indicators for breast cancer and BRCA1/2-mutant breast cancer, and provide new targets and strategies for personalized therapy.

Conclusion

By bioinformatic analyses including GSEA enrichment analysis (GO and KEGG), differentially expressed genes identification, PPI network, survival and diagnostic value analysis, we identified CCNE1, TTK and EXO1 might act as the potential diagnostic indicators for screening BC and BRCA1/2-mutant BC. Our results revealed that cell cycle regulation, cell division and proliferation may play crucial roles in BRCA1/2 mutation BC. A total of 43 overlapping DEGs might play critical functions in the oncogenesis and progression of BRCA1/2-mutant BC, reflected by their specifically changed expression levels in BRCA mutant carriers compared with wild-type BC. Also, CCNE1 and TTK might serve as prognostic biomarkers for BC. However, further validation by molecular biological experiments are required to confirm our investigation. Additional findings obtained in our study (other changed genetic molecules) are also worthy further research.

Supplemental Information

Supplemental Information 1 More raw measurements in our study, including grouping information, sample ID from TCGA dataset and enrichment analysis

Click here for additional data file.

Table S1 Differentially expressed genes in three comparisons

The overlapping genes in three independent DEGs identifications by Venn analysis.

Click here for additional data file.

Supplemental Information 2 Figure S1 and S2

Click here for additional data file.

We are grateful for the data provided by the TCGA database for this study.

Abbreviations

BRCA1/2 Breast cancer susceptibility gene 1/2

BC breast cancer

TNBC Triple-negative breast cancer

TCGA The Cancer Genome Atlas

DEGs Differentially expressed genes

PPI Protein–protein interaction

GSEA Gene set enrichment analysis

KEGG Kyoto Encyclopedia of Genes and Genomes

CNV copy number variation

GO Gene Ontology

BP biological process

FDR false discovery rate

FC fold change

ROC The receiver operating characteristic curve

AUC the area under the curve

ER Estrogen receptor

PR Progesterone receptor

HER2 Human epidermal growth factor receptor 2

OS Overall survival

mTORC1 mechanistic target of rapamycin complex 1

ALB serum albumin

CDKN1A Cyclin-dependent kinase inhibitor 1

CCNE1 cyclin E1

MYOM2 myomesin-2

KRT20 keratin 20

NPBWR1 Neuropeptides B/W receptor 1

SLC4A4 Solute Carrier Family 4 member 4

MAOA Monoamine oxidase A

NLRP7 NLR family pyrin domain containing 7

C4orf51 Chromosome 4 open reading frame 51

A2ML1 Alpha-2-macroglobulin like 1

TTK TTK protein kinase or dual specificity protein kinase

Additional Information and Declarations

Competing Interests

Author Contributions

Data Availability

The authors declare there are no competing interests.

Yue Li conceived and designed the experiments, analyzed the data, prepared figures and/or tables, authored or reviewed drafts of the paper, and approved the final draft.

Xiaoyan Zhou, Ruihua Yang and Qi Wang performed the experiments, prepared figures and/or tables, and approved the final draft.

Jiali Liu performed the experiments, authored or reviewed drafts of the paper, and approved the final draft.

Yang Yin analyzed the data, prepared figures and/or tables, and approved the final draft.

Xiaohong Yuan analyzed the data, authored or reviewed drafts of the paper, and approved the final draft.

Jing Ji analyzed the data, prepared figures and/or tables, and approved the final draft.

Qian He conceived and designed the experiments, authored or reviewed drafts of the paper, and approved the final draft.

The following information was supplied regarding data availability:

Raw measurements are available in the Supplemental Files.

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
