# Peer review of "Differentially expressed genes and key molecules of BRCA1/2-mutant breast cancer: evidence from bioinformatics analyses"

_PeerJ, doi:10.7717/peerj.8403_

## Round 0.1 · original submission · Major Revisions

Please make sure that all the concerns of both reviewers are addressed and the manuscript is revised accordingly.

·

Basic reporting

The article is clear and most of the time direct to the point. The authors used professional, clear and unambiguous, English language. However, occasionally there were some redundant words as indicated in the general comments. Most of the references were relevant to the topic except in few occasions. The authors used a good article structure which includes the relevant figures and tables. But some of the figures were not in a high resolution. In addition, the legend for figure 5 has insufficient description. The results were relevant to the original hypothesis nevertheless; the hypothesis is not very clear in the introduction. Background in the abstract needs more details as well as the introduction needs more details. I suggest adding descriptions in line 21-22 in the abstract and in the line 68-73 in the introduction. Also it is better to write more about the justification for the study and demonstrate adequate relevant literature.

Experimental design

The Peer Journal’s aim and scope is publishing articles in the Biological Sciences, Environmental Sciences, Medical Sciences, and Health Sciences. The current manuscript is a part of the Medical Sciences, and Health Sciences. Therefore, the research is within the aim and scope of the peer journal. The research is defined and meaningful, as well as it is relevant and adding to the scientific knowledge. However, I think the data analysis needs more elaboration. The research is bioinformatics based analysis and most of the used software are well defined. There were no human materials used in this study. So the study in line with the ethical standard. The methods were not described in details and it needs more elaboration and description. The over all data have some degree of novelty.

Validity of the findings

Most of the underlying data were robust and controlled. But most of the figures are not in a very high resolution form. I suggest using a higher resolution form or improving the resolution, especially for the Kaplan-Meier figures. There is no clear conclusion at the end of the article which can summarize the findings.

Additional comments

Abstract:
Line 21. The background in the abstract is insufficient. I suggest to add one or two sentences to reflect on the subject.

Line 37, remove the words “For example”.
Introduction
• Line 64, remove the word mainly.
• Line 66, the sentence “In fact, this limits the opportunity of prevention for BRCA1/2-mutant breast cancer and other tumors such as breast cancer”, it is better to add other tumors.
• Line 70. the sentence “treatment of breast cancer, especially for some hereditary breast cancer and sporadic breast cancer…) is repetitive with redundant words. It is better to change as “treatment of BC, especially for some hereditary and sporadic BC….)
• Line 73, it is better to add a reference.

Methods
• Line 89. No mention for the source of other groups (WT& Control).
• Line 123, the sentence “specializes survival analysis” needs to be cleared. specializes for survival analysis

Results
• Line 128 there is no need to repeat the abbreviation “breast cancer (BC)”
• Line 131, be consistent in writing the BRCA1/2 “BRCA1&BRCA2 mutations”
• Line 162, use the Italian uppercase letter for the BRCA1/2-

Discussion
• Line 212, the sentence “TNBC accounts for about 70% and 20% of BRCA1 and BRCA2 mutation carriers” is in accurate so please review this statement
• Line 255, Italian CCNE1
• Line 273 the sentence “has attracted a lot of research enthusiasm. reported that TTK could regulate the growth” doesn’t make sense.

Reviewer 2 ·

Basic reporting

no comments

Experimental design

no comment

Validity of the findings

no comment

Additional comments

Major Comments:
1) In the methodology section, can the authors elaborate on the ethnic sub-division of the control, Wt and Mutant breast cancer RNA-Seq data? The distinction is important to understand because as authors already mentioned that there is "paucity in data pertaining to ethnical risks with BRCA1/2". Further it is important to observe if the ethnic classification lowers the "N" in differential analysis following inclusion and exclusion criteria. It is also important to know if the authors have performed the PPI integration of differentially expressed genes within the ethnical groups of control vs Wt vs MUT, if it was not done can authors justify the exclusion, especially when it is known to have BRCA1/2 has ethnic risk factors associated with it

2)Could authors provide more detailed description of the RNA-seq data they have obtained from cbiportal? Was it single cell RNA-seq data?How many reads were done on the cells? what was the quality of the data? what kind of algorithm was used for the comparative transcriptome analysis? Was the data obtained from several different research labs? what were the statistical common factors amongst the digital gene expression of the RNA-seq data? what was the method of collection and what measures were undertaken to prevent batch effects and the normalization procedures for the data obtained from different sources?

3)By authors confession in discussion section "lack of adequate BRCA1/2 mutant cases for accurate survival" it is important to further know the statistical confidence level of the predictive model that authors have laid in the manuscript. Can authors comment on the what percentage of confidence do they exude in utilizing the CCNE1, NPBWR1 and TTK genes in prognosis and diagnosis of breast cancer and if that confidence level is enough for a clinical prediction?

4)The genes such as TTK and GRP7 that authors claim to serve as diagnostic markers for BRCA1 breast cancer have previously been studied in the incidence of breast cancer as clinical targets, considering what is already known about these genes/protein products in breast cancer can author comment/defend the merit and novelty of the current study? and how does the study differ from the plethora of literature already available?

Minor comments:
1) Authors please check for some grammatical errors

---

## Round 0.2 · accepted · Accept

The methods and experiments section have been improved. Most of the highlighted comments have been addressed. The article now is acceptable.

·

Basic reporting

I re-reviewed the article. It has been improved and the English language also has been improved greatly. Most of the highlighted comments have been addressed. The article now is ready to be published.

Experimental design

The methods and experiments section have been improved. Most of the highlighted comments have been addressed. The article now is ready to be published.

Validity of the findings

The article has been improved and the English language also has been improved greatly. Most of the highlighted comments have been addressed. The article now is ready to be published.

Additional comments

The methods and experiments section have been improved. Most of the highlighted comments have been addressed. The article now is ready to be published.